# Tango 2: Aligning Diffusion-based Text-to-Audio Generative Models through Direct Preference Optimization

## ABSTRACT

Generative multimodal content is increasingly prevalent in much of the content creation arena, as it has the potential to allow artists and media personnel to create pre-production mockups by quickly bringing their ideas to life. The generation of audio from text prompts is an important aspect of such processes in the music and film industry. Many of the recent diffusion-based text-to-audio models focus on training increasingly sophisticated diffusion models on a large set of datasets of prompt-audio pairs. These models do not explicitly focus on the presence of concepts or events and their temporal ordering in the output audio with respect to the input prompt. Our hypothesis is focusing on how these aspects of audio generation could improve audio generation performance in the presence of limited data. As such, in this work, using an existing text-to-audio model Tango, we synthetically create a preference dataset where each prompt has a winner audio output and some loser audio outputs for the diffusion model to learn from. The loser outputs, in theory, have some concepts from the prompt missing or in an incorrect order. We fine-tune the publicly available Tango text-to-audio model using diffusion-DPO (direct preference optimization) loss on our preference dataset and show that it leads to improved audio output over Tango and AudioLDM2, in terms of both automatic- and manual evaluation metrics.

## CCS CONCEPTS

• **Computing methodologies** → **Natural language processing**;
• **Information systems** → **Multimedia information systems**.

## KEYWORDS

Multimodal AI, Text-to-Audio Generation, Diffusion Models, Large Language Models, Preference Optimization

## 1 INTRODUCTION

Generative AI is increasingly turning into a mainstay of our daily lives, be it directly through using ChatGPT [24], GPT-4 [23] in an assistive capacity, or indirectly by consuming AI-generated memes, generated using models like StableDiffusion [27], DALL-E 3 [1, 22], on social media platforms. Nonetheless, there is a massive demand for AI-generated content across industries, especially in the multimedia sector. Quick creation of audio-visual content or prototypes would require an effective text-to-audio model along

*ACM MM, 2024, Melbourne, Australia*
© 2024 Copyright held by the owner/author(s). Publication rights licensed to ACM.
ACM ISBN 978-x-xxxx-xxxx-x/YY/MM
https://doi.org/10.1145/nnnnnnn.nnnnnnn

with text-to-image and -video models. Thus, improving the fidelity of such models with respect to the input prompts is paramount.

Recently, supervised fine-tuning-based direct preference optimization [26] (DPO) has emerged as a cheaper and more robust alternative to reinforcement learning with human feedback (RLHF) to align LLM responses with human preferences. This idea is subsequently adapted for diffusion models by Wallace et al. [32] to align the denoised outputs to human preferences. In this work, we employ this DPO-diffusion approach to improve the semantic alignment between input prompt and output audio of a text-to-audio model. Particularly, we fine-tune the publicly available text-to-audio latent diffusion model Tango [5] on our synthesized preference dataset with DPO-diffusion loss. This preference dataset contains diverse audio descriptions (*prompts*) with their respective preferred (*winner*) and undesirable (*loser*) audios. The preferred audios are supposed to perfectly reflect their respective textual descriptions, whereas the undesirable audios have some flaws, such as some missing concepts from the prompt or in an incorrect temporal order or high noise level. To this end, we perturbed the descriptions to remove or change the order of certain concepts and passed them to Tango to generate undesirable audios. Another strategy that we adopted for undesirable audio generation was adversarial filtering: generate multiple audios from the original prompt and choose the audio samples with CLAP-score below a certain threshold. We call this preference dataset `Audio-alpaca`. To mitigate the effect of noisy preference pairs stemming from automatic generation, we further choose a subset of samples for DPO fine-tuning based on certain thresholds defined on the CLAP-score differential between preferred and undesirable audios and the CLAP-score of the undesirable audios. This likely ensures a minimal proximity to the input prompt, while guaranteeing a minimum distance between the preference pairs.

We experimentally show that fine-tuning Tango on the pruned `Audio-alpaca` yields Tango 2 that significantly surpasses Tango and AudioLDM2 in both objective and human evaluations. Moreover, exposure to the contrast between good and bad audio outputs during DPO fine-tuning likely allows Tango 2 to better map the semantics of the input prompt into the audio space, despite relying on the same dataset as Tango for synthetic preference data-creation.

The broad contributions of this paper are the following:

(1) We develop a cheap and effective heuristics for semi automatically creating a preference dataset for text-to-audio generation;

(2) On the same note, we also share the preference dataset `Audio-alpaca` for text-to-audio generation that may aid in the future development of such models;

(3) Despite not sourcing additional out-of-distribution text-audio pairs over Tango, our model Tango 2 outperforms both Tango and AudioLDM2 on both objective and subjective metrics;

(4) Tango 2 demonstrates the applicability of diffusion-DPO in audio generation.

## 2 RELATED WORK

Text-to-audio generation has garnered serious attention lately thanks to models like AudioLDM [17], Make-an-Audio [9], Tango [5], and Audiogen [14]. These models rely on diffusion architectures for audio generation from textual prompts. Recently, AudioLM [2] was proposed which utilizes the state-of-the-art semantic model w2v-Bert [4] to generate semantic tokens from audio prompts. These tokens condition the generation of acoustic tokens, which are decoded using the acoustic model SoundStream [34] to produce audio. The semantic tokens generated by w2v-Bert are crucial for conditioning the generation of acoustic tokens, subsequently decoded by SoundStream.

AudioLDM [17] is a text-to-audio framework that employs CLAP [33], a joint audio-text representation model, and a latent diffusion model (LDM). Specifically, an LDM is trained to generate latent representations of melspectrograms obtained using a Variational Autoencoder (VAE). During diffusion, CLAP embeddings guide the generation process. Tango [6] utilizes the pre-trained VAE from AudioLDM and replaces the CLAP model with a fine-tuned large language model: FLAN-T5. This substitution aims to achieve comparable or superior results while training with a significantly smaller dataset.

In the realm of aligning generated audio with human perception, Liao et al. [16] recently introduced BATON, a framework that initially gathers pairs of audio and textual prompts, followed by annotating them based on human preference. This dataset is subsequently employed to train a reward model. The reward generated by this model is then integrated into the standard diffusion loss to guide the network, leveraging feedback from the reward model. However, our approach significantly diverges from this work in two key aspects: 1) we automatically construct a *pairwise* preference dataset, referred to as Audio-alpaca, utilizing various techniques such as LLM-guided prompt perturbation and re-ranking of generated audio from Tango using CLAP scores, and 2) we then train Tango on Audio-alpaca using diffusion-DPO to generate audio samples preferred by human perception.

## 3 BACKGROUND

### 3.1 Overview of Tango

Tango, proposed by Ghosal et al. [5], primarily relies on a latent diffusion model (LDM) and an instruction-tuned LLM for text-to-audio generation. It has three major components:

(1) Textual-prompt encoder
(2) Latent diffusion model (LDM)
(3) Audio VAE and Vocoder

The textual-prompt encoder encodes the input description of the audio. Subsequently, the textual representation is used to construct a latent representation of the audio or audio prior from standard Gaussian noise, using reverse diffusion. Thereafter, the decoder of the mel-spectrogram VAE constructs a mel-spectrogram from the latent audio representation. This mel-spectrogram is fed to a vocoder to generate the final audio.

*3.1.1 Textual Prompt Encoder.* Tango utilizes the pre-trained LLM Flan-T5-Large (780M) [3] as the text encoder ($E_{text}$) to acquire text encoding $\tau \in \mathbb{R}^{L \times d_{text}}$, where $L$ and $d_{text}$ represent the token count and token-embedding size, respectively.

*3.1.2 Latent Diffusion Model.* For ease of understanding, we briefly introduce the LDM of Tango in this section. The latent diffusion model (LDM) [27] in Tango is derived from the work of Liu et al. [18], aiming to construct the audio prior $x_0$ guided by text encoding $\tau$. This task essentially involves approximating the true prior $q(x_0|\tau)$ using parameterized $p_\theta(x_0|\tau)$.

LDM achieves this objective through forward and reverse diffusion processes. The forward diffusion represents a Markov chain of Gaussian distributions with scheduled noise parameters $0 < \beta_1 < \beta_2 < \cdots < \beta_N < 1$, facilitating the sampling of noisier versions of $x_0$:

$$q(x_n|x_{n-1}) = \mathcal{N}(\sqrt{1-\beta_n}x_{n-1}, \beta_n \mathbf{I}), \quad (1)$$

$$q(x_n|x_0) = \mathcal{N}(\sqrt{\overline{\alpha}_n}x_0, (1-\overline{\alpha}_n)\mathbf{I}), \quad (2)$$

where $N$ is the number of forward diffusion steps, $\alpha_n = 1 - \beta_n$, and $\overline{\alpha}_n = \prod_{i=1}^{n} \alpha_n$. Song et al. [29] show that Eq. (2) conveniently follows from Eq. (1) through reparametrization trick that allows direct sampling of any $x_n$ from $x_0$ via a non-Markovian process:

$$x_n = \sqrt{\overline{\alpha}_n}x_0 + (1-\overline{\alpha}_n)\epsilon, \quad (3)$$

where the noise term $\epsilon \sim \mathcal{N}(\mathbf{0}, \mathbf{I})$. The final step of the forward process yields $x_N \sim \mathcal{N}(\mathbf{0}, \mathbf{I})$.

The reverse process denoises and reconstructs $x_0$ through text-guided noise estimation ($\hat{\epsilon}_\theta$) using loss

$$\mathcal{L}_{LDM} = \sum_{n=1}^{N} \gamma_n \mathbb{E}_{\epsilon_n \sim \mathcal{N}(\mathbf{0},\mathbf{I}),x_0} ||\epsilon_n - \hat{\epsilon}_\theta^{(n)}(x_n, \tau)||_2^2, \quad (4)$$

where $x_n$ is sampled according to Eq. (3) using standard normal noise $\epsilon_n$, $\tau$ represents the text encoding for guidance, and $\gamma_n$ denotes the weight of reverse step $n$ [8], interpreted as a measure of signal-to-noise ratio (SNR) relative to $\alpha_{1:N}$. The estimated noise is then employed for the reconstruction of $x_0$:

$$p_\theta(x_{0:N}|\tau) = p(x_N) \prod_{n=1}^{N} p_\theta(x_{n-1}|x_n, \tau), \quad (5)$$

$$p_\theta(x_{n-1}|x_n, \tau) = \mathcal{N}(\mu_\theta^{(n)}(x_n, \tau), \tilde{\beta}^{(n)}), \quad (6)$$

$$\mu_\theta^{(n)}(x_n, \tau) = \frac{1}{\sqrt{\alpha_n}}[x_n - \frac{1-\alpha_n}{\sqrt{1-\overline{\alpha}_n}}\hat{\epsilon}_\theta^{(n)}(x_n, \tau)], \quad (7)$$

$$\tilde{\beta}^{(n)} = \frac{1-\bar{\alpha}_{n-1}}{1-\bar{\alpha}_n}\beta_n. \quad (8)$$

The parameterization of noise estimation $\hat{\epsilon}_\theta$ involves utilizing U-Net [28], incorporating a cross-attention component to integrate the textual guidance $\tau$.

*3.1.3 Audio VAE and Vocoder.* The audio variational auto-encoder (VAE) [11] compresses the mel-spectrogram of an audio sample, $m \in \mathbb{R}^{T \times F}$, into an audio prior $x_0 \in \mathbb{R}^{C \times T/r \times F/r}$, where $C$, $T$, $F$, and $r$ denote the number of channels, time-slots, frequency-slots, and compression level, respectively. The latent diffusion model (LDM) reconstructs the audio prior $\hat{x}_0$ using input-text guidance $\tau$. Both

the encoder and decoder consist of ResUNet blocks [13] and are trained by maximizing the evidence lower-bound (ELBO) [11] and minimizing adversarial loss [10]. Tango utilizes the checkpoint of the audio VAE provided by Liu et al. [18].

As a vocoder to convert the audio-VAE decoder-generated mel-spectrogram into audio, Tango employs HiFi-GAN [12] which is also utilized by Liu et al. [18].

Finally, Tango utilizes a data augmentation method that merges two audio signals while considering human auditory perception. This involves computing the pressure level of each audio signal and adjusting the weights of the signals to prevent the dominance of the signal with higher pressure level over the one with lower pressure level. Specifically, when fusing two audio signals, the relative pressure level is computed using the following equation:

$$p = (1 + 10^{\frac{G_1 - G_2}{20}})^{-1}, \tag{9}$$

Here $G_1$ and $G_2$ are the pressure levels of signal $x_1$ and $x_2$. Then the audio signals are mixed using the equation below:

$$\text{mix}(x_1, x_2) = \frac{px_1 + (1 - p)x_2}{\sqrt{p^2 + (1 - p)^2}}. \tag{10}$$

The denominator is to account for the fact that the energy of a sound wave is proportional to the square of its amplitude as shown in Tokozume et al. [30]. Note that in this augmentation, textual prompts are also concatenated.

## 3.2 Preference Optimization for Language Models

Tuning Large Language Models (LLMs) to generate responses according to human preference has been a great interest to the ML community. The most popular approach for aligning language models to human preference is reinforcement learning with human feedback (RLHF). It comprises the following steps [26]:

**Supervised Fine Tuning (SFT).** First, the pre-trained LLM undergoes supervised fine-tuning on high-quality downstream tasks to obtain the fine-tuned model $\pi^{SFT}$.

**Reward Modeling.** Next, $\pi^{SFT}$ is prompted with an input $\tau$ to generate multiple responses. These responses are then shown to human labelers to rank. Once such a rank is obtained, $x^w \succ x^l \mid \tau$ indicating $x^w$ is preferred over $x^l$, the task is to model these preferences. Among several popular choices of preference modeling, Bradley-Terry (BT) is the most popular one which relies on the equation below:

$$p^*(x^w \succ x^l \mid \tau) = \frac{\exp(r^*(\tau, x^w))}{\exp(r^*(\tau, x^w)) + \exp(r^*(\tau, x^l))} \tag{11}$$

The overall idea is to learn the human preference distribution $p^*$. $r^*(\tau, x)$ is a latent reward function that generates the preferences. With a static dataset created by human annotators, $\mathcal{D} = \left\{ \left( \tau_{(i)}, x_{(i)}^w, x_{(i)}^l \right) \right\}_{i=1}^N$, one can train a reward model $r_\phi(\tau, x)$ using maximum likelihood estimation. The negative log-likelihood loss of this training can be written as follows:

$$\mathcal{L}_R(r_\phi, \mathcal{D}) = -\mathbb{E}_{(\tau, x^w, x^l) \sim \mathcal{D}} \left[ \log \sigma(r_\phi(\tau, x^w) - r_\phi(\tau, x^l)) \right] \tag{12}$$

This formulation considers framing the problem as a binary classification problem.

**RL Optimization.** The final step is to leverage $r_\phi(\tau, x)$ to feedback the language model. As explained by Rafailov et al. [26], this can be embedded into the following learning objective:

$$\max_{\pi_\theta} \mathbb{E}_{\tau \sim \mathcal{D}, x \sim \pi_\theta(x|\tau)} \left[ r_\phi(\tau, x) \right] - \beta D_{KL} \left[ \pi_\theta(x|\tau) \parallel \pi_{\text{ref}}(x|\tau) \right] \tag{13}$$

Here, $\pi_{\text{ref}}$ represents the reference model, which in this context is the supervised fine-tuned model denoted as $\pi^{SFT}$. $\pi_\theta$ stands for the policy language model, intended for enhancement based on feedback from $r_\phi(\tau, x)$. $\beta$ governs $\pi_\theta$ to prevent significant divergence from $\pi_{\text{ref}}$. This control is crucial as it ensures that the model stays close to the distributions upon which $r_\phi(\tau, x)$ was trained. Since the outputs from LLM are discrete, Eq. (13) becomes non-differentiable, necessitating reinforcement learning methods like PPO to address this objective.

## 4 METHODOLOGY

The two major parts of our approach (i) creation of preference dataset Audio-alpaca and (ii) DPO for alignment are outlined in Fig. 1.

### 4.1 Creation of Audio-alpaca

*4.1.1 Audio Generation from Text Prompts.* Our first step is to create audio samples from various text prompts with the pre-trained Tango model. We follow three different strategies as follows:

**Strategy 1: Multiple Inferences from the same Prompt.** In the first setting, we start by selecting a subset of diverse captions from the training split of the AudioCaps dataset. We use the sentence embedding model gte-large[1] [15] to compute dense embedding vectors of all the captions in the training set. We then perform K-Means clustering on the embedded vectors with 200 clusters. Finally, we select 70 samples from each cluster to obtain a total of 14,000 captions. We denote the selected caption set as $\mathcal{T}_1$.

The captions selected through the above process constitute the seed caption set. Now, we follow two settings to generate audio samples from these captions:

(1) **Strategy 1.1**: Prompt Tango-full-FT with the caption to generate four different audio samples with 5, 25, 50, and 100 denoising steps. All samples are created with a guidance scale of 3.

(2) **Strategy 1.2**: Prompt Tango-full-FT with the caption to generate four different audio samples each with 50 denoising steps. All samples are created with a guidance scale of 3.

In summary, we obtain $(\tau, x_1, x_2, x_3, x_4)$ from **Strategy 1**, where $\tau$ denotes the caption from $\mathcal{T}_1$ and $x_i$ denotes the audios generated from $\tau$.

**Strategy 2: Inferences from Perturbed Prompts.** We start from the selected set $\mathcal{T}_1$ and make perturbations of the captions using the GPT-4 language model [23]. For a caption $\tau$ from $\mathcal{T}_1$, we denote $\tau_1$ as the perturbed caption generated from GPT-4. We add

---

[1]hf.co/thenlper/gte-large

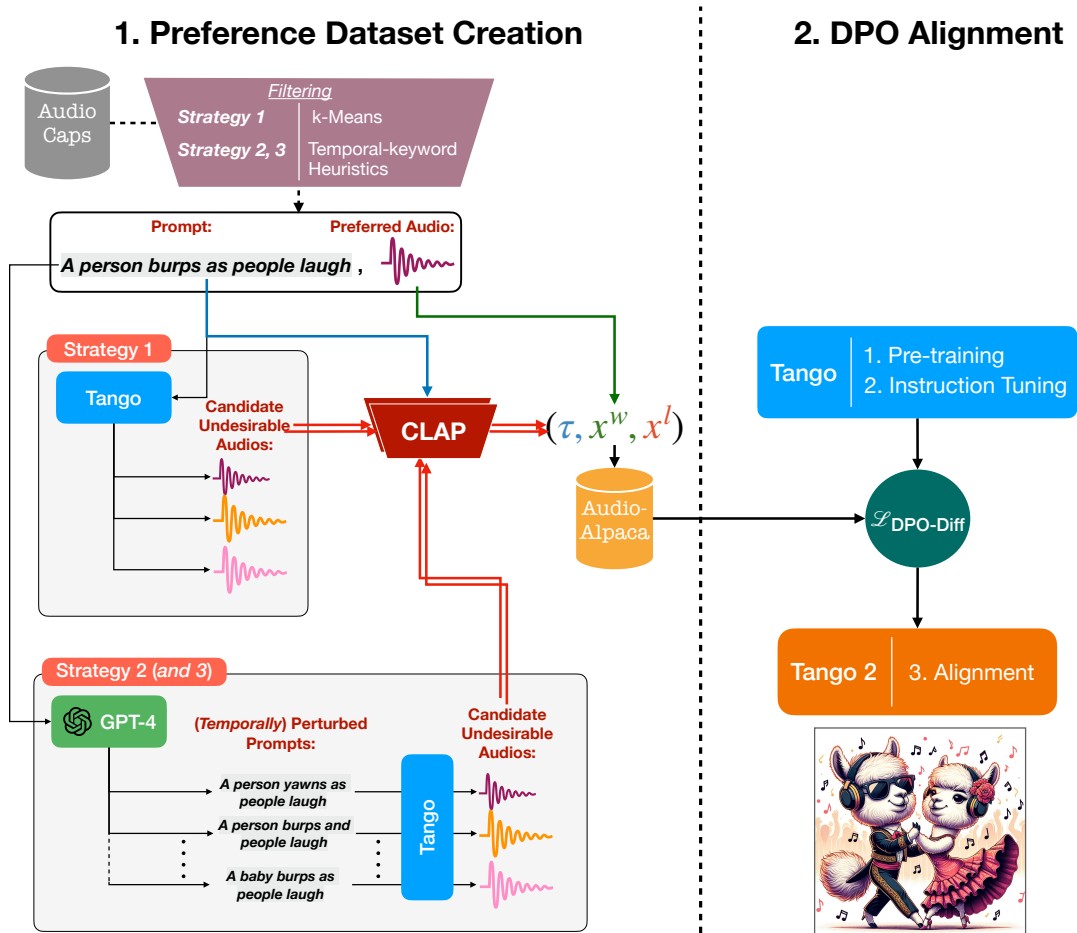

**Figure 1: An illustration of our pipeline for text-to-audio alignment. The left part depicts the preference dataset creation where three strategies are deployed to generate the undesirable audio outputs to the input prompts. These samples are further filtered to form `Audio-alpaca`. A part of this preference dataset is finally used to align TANGO using DPO-diffusion loss (Eq. (17)), resulting TANGO 2.**

specific instructions in our input prompt to make sure that $\tau_1$ is semantically or conceptually close to $\tau$. We show an illustration of the process in Table 1. In practice, we create five different perturbed $\tau_1$ for each $\tau$ from GPT-4, as shown in Table 1.

We then prompt Tango-full-FT with $\tau$ and $\tau_1$ to generate audio samples $x_\tau$ and $x_{\tau_1}$. We use 50 denoising steps with a guidance scale of 3 to generate these audio samples.

To summarize, we obtain $(\tau, x_\tau, x_{\tau_1})$ from **Strategy 2**. Note that, we considered $\tau_1$ only to generate the audio sample $x_{\tau_1}$. We do not further consider $\tau_1$ while creating the preference dataset.

**Strategy 3: Inferences from Temporally Perturbed Prompts.** This strategy is aimed at prompts that describe some composition of sequence and simultaneity of events. To identify such prompts in `AudioCaps`' training dataset, as a heuristics, we look for the following keywords in a prompt: *before*, *after*, *then*, or *followed*. We denote the set of such prompts as $\mathcal{T}_2$.

For each caption $\tau_2$ in $\mathcal{T}_2$, we then prompt GPT-4 to create a set of temporal perturbations. The temporal perturbations include changing the order of the events in the original caption, or introducing a new event or removing an existing event, etc. We aim to create these temporal perturbations by providing specific instructions to GPT-4, which we also illustrate in Table 1.

We denote the temporally perturbed caption as $\tau_2$. We then follow the same process as mentioned earlier in *Strategy 2* to create the audio samples $x_\tau$ and $x_{\tau_2}$. Finally, we pair the $(\tau, x_\tau, x_{\tau_2})$ samples from this strategy. Analogous to the previous strategy, the $\tau_2$ is only used to create the $x_{\tau_2}$, and is not used anywhere else for preference data creation.

We collect the paired text prompt and audio samples from the three strategies and denote it overall as $(\tau, \langle x \rangle)$, where $\langle x \rangle$ indicates the set of 4 or 2 generated audio samples depending upon the corresponding strategy.

*4.1.2 Ranking and Preference-Data Selection.* We first create a pool of candidate preference data for the three strategies as follows:

**Table 1: Prompts used in GPT-4 for creating the perturbations and the corresponding output.**

| *Strategy* and Original Caption | Prompt to GPT-4 | Output from GPT-4 |
|---|---|---|
| *Strategy 2: Perturbed Prompts* 

 People cheering and race cars racing by. | I have an audio clip for which the original caption is as follows: People cheering and race cars racing by. Can you generate five candidate captions that would satisfy the following requirements: 
 i) Would be closely related to the audio clip and the original caption. 
 ii) Would contain inaccuracies in terms of describing the audio clip i.e. they would be somewhat wrong captions for the audio clip. 
 iii) However, the new captions should not be completely unrelated. Always keep some concepts from the original caption in the new one. 
 iv) Would be of similar length to the original caption. 
 Generate only the captions in separate lines so that I can programmatically extract them later. | 1. Crowd applauding and bicycles speeding past. 

 2. Spectators clapping and trains rushing by. 
 3. Audience cheering and horses galloping past. 

 4. Fans shouting and airplanes flying by. 

 5. Group celebrating and motorcycles revving past. |
| *Strategy 3: Temporally-Perturbed Prompts* 

 A man is speaking then a sewing machine briefly turns on and off. | I have an audio clip for which the original caption is as follows: A man is speaking then a sewing machine briefly turns on and off. Can you generate five candidate captions that would satisfy the following requirements: 
 i) Would be closely related to the audio clip and the original caption. 

 ii) Would have a change of order of the events described in the original caption. 
 iii) Would contain inaccuracies in terms of describing the audio clip i.e. they would be somewhat wrong captions for the audio clip. 
 iv) However, the new captions should not be completely unrelated. Always keep some concepts from the original caption in the new one. 
 v) Would be of similar length to the original caption. 
 Generate only the captions in separate lines so that I can programmatically extract them later. | 1. A sewing machine briefly starts, then a man begins speaking. 

 2. The sound of a sewing machine is heard after a man's voice. 

 3. A man's voice is followed by the noise of a blender. 

 4. A woman speaks and then a sewing machine is turned on. 

 5. The noise of a sewing machine is interrupted by a man talking. |

**For Strategy 1.** Let's assume we have an instance $(\tau, \langle x \rangle)$ from Strategy 1. We first compute the CLAP matching score following Wu et al. [33] between $\tau$ and all the four audio samples in $\langle x \rangle$. We surmise that the sample in $\langle x \rangle$ that has the highest matching score with $\tau$ is most aligned with $\tau$, compared to the other three audio samples that have a relatively lower matching score. We consider this audio with the highest matching score as the winning sample $x^w$ and the other three audio samples as the losing sample $x^l$. In this setting, we can thus create a pool of three preference data points: $(\tau, x^w, x^l)$, for the three losing audio samples $x^l$.

**For Strategy 2 and 3.** Let's assume we have an instance $(\tau, \langle x \rangle)$ from Strategy 2 or 3. We compute the CLAP matching score between i) $\tau$ with $x_\tau$, and ii) $\tau$ with the $x_{\tau_1}$ or $x_{\tau_2}$, corresponding to the strategy. We consider only those instances where the CLAP score of i) is higher than the CLAP score of ii). For these instances, we use $x_\tau$ as the winning audio $x^w$ and $x_{\tau_1}$ or $x_{\tau_2}$ as the losing audio $x^l$ to create the preference data point: $(\tau, x^w, x^l)$.

**Final Selection.** We want to ensure that the winning audio sample $x^w$ is strongly aligned with the text prompt $\tau$. At the same time, the winning audio sample should have a considerably higher alignment with the text prompt than the losing audio sample. We use the CLAP score as a measurement to fulfill these conditions. The CLAP score is measured using cosine similarity between the text and audio embeddings, where higher scores indicate higher alignment between the text and the audio. We thus use the following conditions to select a subset of instances from the pool of preference data:

(1) The winning audio must have a minimum CLAP score of $\alpha$ with the text prompt to ensure that the winning audio is strongly aligned with the text prompt.
(2) The losing audio must have a minimum CLAP score of $\beta$ with the text prompt to ensure that we have semantically close negatives that are useful for preference modeling.

(3) The winning audio must have a higher CLAP score than the losing audio w.r.t to the text prompt.
(4) We denote $\Delta$ to be the difference in CLAP score between the text prompt with the winning audio[2] and the text prompt with the losing audio. The $\Delta$ should lie between certain thresholds, where the lower bound will ensure that the losing audio isn't too close to the winning audio, and the upper bound will ensure that the losing audio is not too undesirable.

We use an *ensemble filtering* strategy based on two different CLAP models: 630k-audioset-best and 630k-best [33]. This can reduce the effect of noise from individual CLAP checkpoints and increase the robustness of the selection process. In this strategy, samples are included in our preference dataset if and only if they satisfy all the above conditions based on CLAP scores from both of the models. We denote the conditional scores mentioned above as $\alpha_1, \beta_1, \Delta_1$, and $\alpha_2, \beta_2, \Delta_2$ for the two CLAP models, respectively. Based on our analysis of the distribution of the CLAP scores as shown in Figure 2, we choose their values as follows: $\alpha_1 = 0.45$, $\alpha_2 = 0.60$, $\beta_1 = 0.40$, $\beta_2 = 0.0$, $0.05 \leq \Delta_1 \leq 0.35$, and $0.08 \leq \Delta_2 \leq 0.70$.

Finally, our preference dataset `Audio-alpaca` has a total of $\approx$ 15k samples after this selection process. We report the distribution of `Audio-alpaca` in Table 2.

## 4.2 DPO for Preference Modeling

As opposed to RLHF, recently DPO has emerged as a more robust and often more practical and straightforward alternative for LLM alignment that is based on the very same BT preference model (Eq. (11)). In contrast with supervised fine-tuning (SFT) that only optimizes for the desirable (*winner*) outputs, the DPO objective also allows the model to learn from undesirable (*loser*) outputs, which is key in the absence of a high-quality reward model, as required for RLHF. To this end, the DPO objective is derived by substituting

---

[2]In our paper, we employ the terms "winner" and "preferred" interchangeably. Likewise, we use "loser" and "undesirable" interchangeably throughout the text.

**Table 2: Statistic of Audio-alpaca**

| Strategy | # Samples | Avg. Winner Score | Avg. Loser Score | Avg. Delta |
|---|---|---|---|---|
| Inference w/ Different Denoising Steps (Strategy 1.1) | 3004 | 0.645 | 0.447 | 0.198 |
| Inference w/ Same Denoising Steps (Strategy 1.2) | 2725 | 0.647 | 0.494 | 0.153 |
| GPT-4 Perturbed Prompts (Strategy 2) | 4544 | 0.641 | 0.425 | 0.216 |
| GPT-4 Temporally Perturbed Prompts (Strategy 3) | 4752 | 0.649 | 0.458 | 0.191 |
| Overall | 15025 | 0.645 | 0.452 | 0.193 |

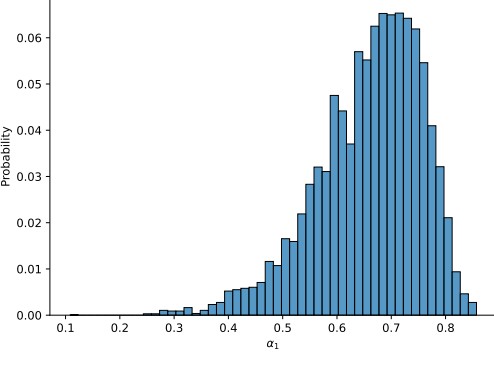

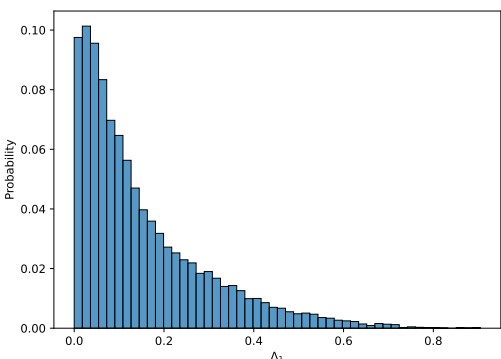

**Figure 2: The distribution of $\alpha_1$ and $\Delta_1$ scores in the unfiltered dataset. We see that for an unfiltered dataset: i) the winner audio sample is not always strongly aligned to the text prompt in the $\alpha_1$ plot; ii) winner and loser audio samples can be too close in the $\Delta_1$ plot. We thus choose the values of our $\alpha_1$, $\Delta_1$ and other selection parameters to ensure the filtered dataset is less noisy with more separation between the winner and loser audios.**

the globally optimal reward—obtained by solving Eq. (13)—in the negative log-likelihood (NLL) loss (Eq. (12)).

This success spurred on Wallace et al. [32] to bring the same benefits of DPO to diffusion networks. However, unlike DPO, the goal for diffusion networks is to maximize the following learning objective (Eq. (14)) with a reward (Eq. (15)) defined on the entire diffusion path $x_{0:N}$:

$$\max_{\pi_\theta} \mathbb{E}_{\tau \sim \mathcal{D}, x_{0:N} \sim \pi_\theta(x_{0:N}|\tau)} [r(\tau, x_0)$$
$$- \beta D_{\mathrm{KL}}[\pi_\theta(x_{0:N}|\tau)||\pi_{\mathrm{ref}}(x_{0:N}|\tau)]. \quad (14)$$
$$r(\tau, x_0) := \mathbb{E}_{\pi_\theta(x_{1:N}|x_0,\tau)}[R(\tau, x_{0:N})], \quad (15)$$

Solving this objective and substituting the optimal reward in the NLL loss (Eq. (12)) yields the following DPO objective for diffusion:

$$\mathcal{L}_{\mathrm{DPO\text{-}Diff}} = -\mathbb{E}_{(\tau, x_0^w, x_0^l) \sim \mathcal{D}_{\mathrm{pref}}} \log \sigma ($$

$$\beta \mathbb{E}_{x_{1:N}^* \sim \pi_\theta(x_{1:N}^*|x_0^*,\tau)} [\log \frac{\pi_\theta(x_{0:N}^w|\tau)}{\pi_{\mathrm{ref}}(x_{0:N}^w|\tau)} - \log \frac{\pi_\theta(x_{0:N}^l|\tau)}{\pi_{\mathrm{ref}}(x_{0:N}^l|\tau)}]). \quad (16)$$

Now, applying Jensen's inequality by taking advantage of the convexity of $-\log \sigma$ allows the inner expectation to be pushed outside. Subsequently, approximating the denoising process with the forward process yields the following final form in terms of the L2 noise-estimation losses from LDM (Eq. (4)):

$$\mathcal{L}_{\mathrm{DPO\text{-}Diff}} := -\mathbb{E}_{n, \epsilon^w, \epsilon^l} \log \sigma(-\beta N \omega(\lambda_n)(||\epsilon_n^w - \hat{\epsilon}_\theta^{(n)}(x_n^w, \tau)||_2^2$$
$$- ||\epsilon_n^w - \hat{\epsilon}_{\mathrm{ref}}^{(n)}(x_n^w, \tau)||_2^2$$
$$- (||\epsilon_n^l - \hat{\epsilon}_\theta^{(n)}(x_n^l, \tau)||_2^2 - ||\epsilon_n^l - \hat{\epsilon}_{\mathrm{ref}}^{(n)}(x_n^l, \tau)||_2^2)), \quad (17)$$

where $\mathcal{D}_{\mathrm{pref}} := \{(\tau, x_0^w, x_0^l)\}$ is our preference dataset Audio-alpaca, $\tau$, $x_0^w$, and $x_0^l$ being the input prompt, preferred, and undesirable output, respectively. Furthermore, $n \sim \mathcal{U}(0, N)$ is the diffusion step, $\epsilon_n^* \sim \mathcal{N}(0, \mathbb{I})$ and $x_n^*$ are noise and noisy posteriors, respectively, at some step $n$. $\lambda_n$ is the signal-to-noise ratio (SNR) and $\omega(\lambda_n)$ is a weighting function defined on SNR. We use Tango-full-FT as our reference model through its noise estimation $\hat{\epsilon}_{\mathrm{ref}}$.

## 5 EXPERIMENTS

### 5.1 Datasets and Training Details

We fine-tuned our model starting from the Tango-full-FT checkpoint on our preference dataset Audio-alpaca. As mentioned earlier in Section 4.1.2, we have a total of 15,025 preference pairs in Audio-alpaca, which we use for fine-tuning. We use AdamW [20] with a learning rate of 9.6e-7 and a linear learning-rate scheduler for fine-tuning. Following Wallace et al. [32], we set the $\beta$ in DPO loss (Eq. (17)) to 2000. We performed 1 epoch of supervised fine-tuning on the prompt and the preferred audio as training samples, followed by 4 epochs of DPO. The entirety of the fine-tuning was executed on two A100 GPUs which takes about 3.5 hours in total. We use a per GPU batch size of 4 and a gradient accumulation step of 4, resulting in an effective batch size of 32.

### 5.2 Baselines

We primarily compare Tango 2 to three strong baselines:

(1) **AudioLDM** [17]: A text-to-audio model that uses CLAP [33], a joint audio-text representation model, and a latent diffusion model (LDM). Specifically, the LDM is trained to generate the

latent representations of melspectrograms obtained from a pre-trained Variational Autoencoder (VAE). During diffusion, CLAP text-embeddings guide the generation process.

(2) **AudioLDM2** [19]: An any-to-audio framework which uses language of audio (LOA) as a joint encoding of audio, text, image, video, and other modalities. Audio modality is encoded into LOA using a self-supervised masked auto-encoder. The remaining modalities, including audio again, are mapped to LOA through a composition of GPT-2 [25] and ImageBind [7]. This joint encoding is used as a conditioning in the diffusion network for audio generation.

(3) **Tango** [5]: Utilizes the pre-trained VAE from AudioLDM but replaces the CLAP text-encoder with an instruction-tuned large language model: FLAN-T5. As compared to AudioLDM, its data-augmentation strategy is also cognizant of the audio pressure levels of the source audios. These innovations attain comparable or superior results while training on a significantly smaller dataset.

Baton [16] represents another recent approach in human preference based text-to-audio modeling. It trains a reward model to maximize rewards through supervised fine-tuning, aiming to maximize the probability of generating audio from a textual prompt. As discussed in Section 2, Baton's reward model is not trained using the pairwise preference objective presented in Equation (12). In this approach, each text ($\tau$) and audio ($x$) pair is classified as 1 or 0, indicating whether human annotators favored the text-audio pair or not. Subsequently, this reward is incorporated into the generative objective function of the diffusion. This methodology stands in contrast to the prevailing approach in LLM alignment research. As of now, neither the dataset nor the code has been made available for comparison.

### 5.3 Evaluation Metrics

**Objective Metrics.** We evaluated the text-to-audio generation using the standard Frechet Audio Distance (FAD), KL divergence, Inception score (IS), and CLAP score [17]. *FAD* is adapted from Frechet Inception Distance (FID) and measures the distribution-level gap between generated and reference audio samples. *KL divergence* is an instance-level reference-dependent metric that measures the divergence between the acoustic event posteriors of the ground truth and the generated audio sample. FAD and KL are computed using PANN, an audio-event tagger. *IS* evaluates the specificity and coverage of a set of samples, not needing reference audios. IS is inversely proportional to the entropy of the instance posteriors and directly proportional to the entropy of the marginal posteriors. *CLAP score* is defined as the cosine similarity between the CLAP encodings of an audio and its textual description. We borrowed the AudioLDM evaluation toolkit [17] for the computation of FAD, IS, and KL scores.

**Subjective Metrics.** Our subjective assessment examines two key aspects of the generated audio: overall audio quality (OVL) and relevance to the text input (REL), mirroring the approach outlined in the previous works, such as, [5, 31]. The OVL metric primarily gauges the general sound quality, clarity, and naturalness irrespective of its alignment with the input prompt. Conversely, the REL metric assesses how well the generated audio corresponds to the

given text input. Annotators were tasked with rating each audio sample on a scale from 1 (least) to 5 (highest) for both OVL and REL. This evaluation was conducted on a subset of 50 randomly-selected prompts from the AudioCaps test set, with each sample being independently reviewed by at least four annotators. Please refer to the supplementary material for more details on the evaluation instructions and evaluators.

### 5.4 Main Results

We report the comparative evaluations of Tango 2 against the baselines Tango [5] and AudioLDM2 [19] in Table 3. For a fair comparison, we used exactly 200 inference steps in all our experiments. Tango and Tango 2 were evaluated with a classifier-free guidance scale of 3 while AudioLDM2 uses a default guidance scale of 3.5. We generate only one sample per text prompt.

**Objective Evaluations.** Tango 2 achieves notable improvements in objective metrics, with scores of 2.69 for FAD, 1.12 for KL, 9.09 for IS, and 0.57 for CLAP. While FAD, KL, and IS assess general naturalness, diversity, and audio quality, CLAP measures the semantic alignment between the input prompt and the generated audio. As documented in Melechovsky et al. [21], enhancing audio quality typically relies on improving the pre-training process of the backbone, either through architectural modifications or by leveraging larger or refined datasets. However, in our experiments, we observe enhanced audio quality in two out of three metrics, specifically KL and IS. Notably, Tango 2 also significantly outperforms various versions of AudioLDM and AudioLDM2 on these two metrics.

On the other hand, we note a substantial enhancement in the CLAP score. CLAP score is particularly crucial in our experimental setup as it directly measures the semantic alignment between the textual prompt and the generated audio. This outcome suggests that DPO-based fine-tuning on the preference data from Audio-alpaca yields superior audio generation to Tango and AudioLDM2.

**Subjective Evaluations.** In our subjective evaluation, Tango 2 achieves high ratings of 3.99 in OVL (overall quality) and 4.07 in REL (relevance), surpassing both Tango and AudioLDM2. This suggests that Tango 2 significantly benefits from preference modeling on Audio-alpaca. Interestingly, our subjective findings diverge from those reported by Melechovsky et al. [21]. In their study, the authors observed lower audio quality when Tango was fine-tuned on music data. However, in our experiments, the objective of preference modeling enhances both overall sound quality and the relevance of generated audio to the input prompts. Notably, in our experiments, AudioLDM2 performed the worst, with the scores of only 3.56 in OVL and 3.19 in REL, significantly lower than both Tango and Tango 2.

### 5.5 Analyses

**Results on Temporal and Multi-concept Prompts.** We analyze the performance of Tango and AudioLDM2 models in audio generation when text prompts contain multiple sequential events. For instance, *"Two gunshots followed by birds flying away then a boy laughing"* consists of three distinct events in a sequence. In contrast, *"A man is snoring"* lacks any temporal sequence. We

**Table 3: Text-to-audio generation results on AudioCaps evaluation set. Due to time and budget constraints, we could only subjectively evaluate AudioLDM 2-Full-Large and Tango-full-FT. Notably these two models are considered open-sourced SOTA models for text-to-audio generation as reported in [31].**

| Model | #Parameters | Objective | | | | Subjective | |
|---|---|---|---|---|---|---|---|
| | | FAD ↓ | KL ↓ | IS ↑ | CLAP ↑ | OVL ↑ | REL ↑ |
| AudioLDM-M-Full-FT | 416M | 2.57 | 1.26 | 8.34 | 0.43 | – | – |
| AudioLDM-L-Full | 739M | 4.18 | 1.76 | 7.76 | 0.43 | – | – |
| AudioLDM 2-Full | 346M | 2.18 | 1.62 | 6.92 | 0.43 | – | – |
| AudioLDM 2-Full-Large | 712M | **2.11** | 1.54 | 8.29 | 0.44 | 3.56 | 3.19 |
| Tango-full-FT | 866M | 2.51 | 1.15 | 7.87 | 0.54 | 3.81 | 3.77 |
| Tango 2 | 866M | 2.69 | **1.12** | **9.09** | **0.57** | **3.99** | **4.07** |

**Table 4: Objective evaluation results for audio generation in the presence of multiple concepts or a single concept in the text prompt in the AudioCaps test set.**

| Model | Multiple Concepts | | | | | | Single Concept | | | | | |
|---|---|---|---|---|---|---|---|---|---|---|---|---|
| | Objective | | | | Subjective | | Objective | | | | Subjective | |
| | FAD ↓ | KL ↓ | IS ↑ | CLAP ↑ | OVL ↑ | REL ↑ | FAD ↓ | KL ↓ | IS ↑ | CLAP ↑ | OVL ↑ | REL ↑ |
| AudioLDM 2-Full | **1.93** | 1.52 | 6.93 | 0.42 | – | – | 2.68 | 2.03 | 6.55 | 0.46 | – | – |
| AudioLDM 2-Full-Large | 2.40 | 1.43 | 7.14 | 0.43 | 3.54 | 3.16 | 2.70 | 1.96 | 6.60 | 0.47 | 3.65 | 3.41 |
| Tango-full-FT | 2.77 | 1.00 | 6.83 | 0.54 | 3.83 | 3.80 | 3.06 | **1.69** | 6.41 | 0.55 | 3.67 | 3.49 |
| Tango 2 | 3.20 | **0.94** | **7.73** | **0.56** | **3.99** | **4.07** | 2.58 | 1.77 | **7.47** | **0.57** | **3.95** | **4.10** |

**Table 5: Objective evaluation results for audio generation in the presence of temporal events or non-temporal events in the text prompt in the AudioCaps test set.**

| Model | Temporal Events | | | | | | Non Temporal Events | | | | | |
|---|---|---|---|---|---|---|---|---|---|---|---|---|
| | Objective | | | | Subjective | | Objective | | | | Subjective | |
| | FAD ↓ | KL ↓ | IS ↑ | CLAP ↑ | OVL ↑ | REL ↑ | FAD ↓ | KL ↓ | IS ↑ | CLAP ↑ | OVL ↑ | REL ↑ |
| AudioLDM 2-Full | **1.95** | 1.71 | 6.37 | 0.41 | – | – | **2.38** | 1.56 | 7.38 | 0.44 | – | – |
| AudioLDM 2-Full-Large | 2.39 | 1.65 | 6.10 | 0.42 | 3.35 | 2.82 | 2.68 | 1.46 | 8.12 | 0.46 | 3.79 | 3.62 |
| Tango-full-FT | 2.55 | 1.16 | 5.82 | 0.55 | 3.83 | 3.67 | 3.04 | **1.15** | 7.70 | 0.53 | 3.78 | 3.88 |
| Tango 2 | 3.29 | **1.07** | **6.88** | **0.58** | **3.92** | **3.99** | 2.84 | 1.16 | **8.62** | **0.55** | **4.05** | **4.16** |

partition the AudioCaps test set based on temporal identifiers such as "while," "as," "before," "after," "then," and "followed," creating two subsets: one with multiple sequential events and the other with no temporality. Our objective evaluation results for these subsets are shown in Table 5. Tango 2 achieves the highest scores of KL, CLAP, and IS on both temporal and non-temporal events, indicating a consistent trend of performance improvement. Similar trends are observed in subjective evaluations. We attribute this improvement to the augmentation of both temporal and non-temporal inputs in constructing Audio-alpaca.

Additionally, we partition the prompts based on the presence of multiple concepts, such as *"A woman speaks while cooking"*. In general, Tango 2 outperforms AudioLDM2 and Tango across most objective and subjective metrics.

## 6 CONCLUSION

In this work, we propose aligning text-to-audio generative models through direct preference optimization. To the best of our knowledge, this represents the first attempt to advance text-to-audio generation through preference optimization. We achieve this by automatically generating a preference dataset using a combination of Large Language Models (LLMs) and adversarial filtering. Our preference dataset, Audio-alpaca, comprises diverse audio descriptions (prompts) paired with their respective preferred (winner) and undesirable (loser) audios. The preferred audios are expected to accurately reflect their corresponding textual descriptions, while the undesirable audios exhibit flaws such as missing concepts, incorrect temporal order, or high noise levels. To generate undesirable audios, we perturb the descriptions by removing or rearranging certain concepts and feed them to Tango. Additionally, we employ adversarial filtering, generating multiple audios from the original prompt and selecting those with CLAP scores below a specified threshold. Subsequently, we align a diffusion-based text-to-audio model, Tango, on Audio-alpaca using DPO-diffusion loss. Our results demonstrate significant performance leap over the previous models, both in terms of objective and subjective metrics. We anticipate that our dataset, Audio-alpaca, and the proposed model, Tango 2, will pave the way for further advancements in alignment techniques for text-to-audio generation.

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
