# OpenReview forum: "Tango 2: Aligning Diffusion-based Text-to-Audio Generative Models through Direct Preference Optimization"
_acmmm.org/ACMMM/2024/Conference — MM2024 Oral_

### Official Review · Reviewer_FpfJ · 2024-05-24

**Rating:** 2
**Confidence:** 3

**Summary:**

This paper focuses on text to audio generation with an emphasis on enhancing multiple concepts and their temporal ordering accuracy by fine-tuning an existing model using Direct Preference Modelling (DPO). It introduces a method for curating a synthetic preference dataset using the Tango model, a SOTA text-to-audio generator.  The dataset consists of pairs of preferred and undesirable audio. The data curation strategies involve multiple inference and re-ranking, as well as generating audio from perturbed prompts compared to the original prompts. After that, Tango is fine-tuned on the preference dataset using DPO for diffusion models proposed by Wallace et al. Experiments demonstrate that the fine-tuned model achieves higher text-to-audio generation quality in varied aspects, including scenarios with multiple concepts and ordering constraints.

**Strengths:**

Based on the experimental results, the proposed Tango2 model achieves competitive text-to-audio performance with existing SOTAs.

**Limitations:**

This paper is overall well proposed but the reviewer has some questions/concerns as follows:

* Line 801 states “AudioLDM2 … significantly lower than both Tango and Tango2.” Although there is indeed a good margin in this case, the statistical significance is not overall tested in the experiments. The author should be more cautious when using such a term.

* It is not clear to the reviewer if the metrics applied in this paper are discriminative enough to the temporal ordering of sound objects. In Table 2 of Appendix, KL and CLAP barely show any difference with or without Strategy 3 (which introduces temporal perturbation in the text prompts). Is it due to inefficiency of Strategy 3, or insensitivity of these metrics to the temporal ordering? Since enhancing temporal ordering accuracy is a key proposal in this paper, the reviewer deems it necessary to investigate this aspect more thoroughly.

* Based on the proposed synthetic dataset and curation methods, are there any more general insights into the fields such as multimedia and generative modelling? To the reviewer’s understanding, the main technical method regarding DPO is completely adopted from Wallace et al. The technical contribution of this paper is seemingly limited.

A few more suggestions for writing:

* Section 3.1 introduces Tango in great detail, which is, however, potentially distracting (in particular, Eq. (9)-(10) seems not relevant at all to this paper). The reviewer recommends delete Section 3.1 or put it to the Appendix. If DPO is a universal approach, then it need not rely on any specific base model.

* The claims in Line 94 and 101 are written in an uncertain tone (using words such as “likely”). It is not encouraged. If it is not an assumption, then it should be validated by experiment.

* Eq. (12), (15), and (16) use undefined variables $\sigma$, $R$ and $x^*$.

**Suitability:**

3

---

### Official Review · Reviewer_GG24 · 2024-05-24

**Rating:** 4
**Confidence:** 4

**Summary:**

This paper construct a text-audio preference alignment dataset and uses DPO to finetune TANGO model to improve the temporal and content-aspect model performance. The paper demonstrate a good pipeline about how to create such a dataset and share both the subjective and objective evaluation results, where TANGO 2 performs better than previous SOTA models.

**Strengths:**

- The introduction of a novel preference dataset, Audio-alpaca, specifically designed for text-to-audio generation.

- This paper also shows that using DPO to improve the performance of pretrained text-to-audio model works well.

**Limitations:**

Pipeline:
- The pipeline strategy 3 mentioned using temporally perturbing to create prompt pairs, but CLAP more focuses on the semantic similarity; from the original paper, there is no evidence that CLAP embedding contains temporal information, which makes the pipeline and ablation in Section 5.5 not convincing. In this case, the paper should consider using some improved version, such as T-CLAP, to update the pipeline.

Evaluation

Actually the evaluation section is my main concern, since reinforcement learning and DPO finetuning is sensitive to the entire pipeline and the evaluation results should show a reasonable improvement to make the method convincing.

- This paper uses CLAP to pick audio samples, therefore the finetuned TANGO model is affected by CLAP's inductive blas. In this case, evaluating the model with CLAP score is inappropriate. We need to realise that CLAP score is more like a biased model instead of oracle, the further evaluation should be conducted.

- In the evaluation section, the author also points out that TANGO 2 "significantly" performs better than TANGO and AudioLDM 2, but the significance test is not performed at the evaluation stage. Is the improvement really significant? I encourage the paper can add a t-test to show the significant level.

- The subjective evaluation can be a good addition to the objective evaluation, especially when the main metric FAD score increases after DPO finetuning. I believe a significance test is still necessary, and from the original paper, OVL and REL score is not a 5-point metric, it is better to re-scale it to 0-100 scale.

Related work

The paper can consider add the literature below:

[1] Cideron, G., Girgin, S., Verzetti, M., Vincent, D., Kastelic, M., Borsos, Z., ... & Agostinelli, A. (2024). MusicRL: Aligning Music Generation to Human Preferences. arXiv preprint arXiv:2402.04229.

**Suitability:**

3

---

### Official Review · Reviewer_JT2h · 2024-05-25

**Rating:** 5
**Confidence:** 3

**Summary:**

This paper proposes TANGO 2, a fine-tuned version of text-to-audio model TANGO using diffusion-DPO.
Based on a carefully curated caption dataset, the author proposes three different strategies to build caption-audio pairs with different perturbation methods.

**Strengths:**

- The paper is very well written and easy to follow
- The paper presents the first work on utilizing diffusion-DPO for audio generation.
- The paper presents a carefully curated audio preference dataset.
- The experiment has shown that the proposed method can greatly benefit the audio quality and text relevance.

**Limitations:**

- The preference dataset is curated based on the CLAP score. Instead of using the proposed method, I’m curious about the model's performance if the author directly used TANGO to generate multiple samples and filter the best result using the CLAP score.
- The most vital limitation of the proposed method is the heavy reliance on the CLAP score during the data curation process. Still, considering the use of the ensembling filtering strategy and strict threshold set, it’s generally acceptable.
- The presentation could benefit from including case studies or a web page for demo purposes.

**Suitability:**

3

---

### Meta-Review · Area_Chair_1koS · 2024-06-29

**Recommendation:** Accept (Oral)
**Confidence:** 4

**Metareview:**

The quality of the paper is good. It is well-written with some original contributions, such as utilizing diffusion-DPO for audio generation, and a carefully curated audio preference dataset. The authors have done a good job to address reviewers' critiques during the rebuttal period. Hope the authors can take the reviewers' comments into consideration in the camera-ready version, if this paper is accepted.